# Impact of Environmental Factors on Short-term Eye Strain Relief during COVID-19 Quarantine: A Pilot Study

**Yihao Lu [1], Jianan Wang [1,\*], Jianhua Chen [1], Yufan Yan [1], Haicong Zeng [1], Baowei Zhang [2], Haohao Ma [2] and Tingli Hu [2]**

[1] College of Forestry and Landscape Architecture, Anhui Agricultural University, Hefei 230036, China
[2] School of Life Sciences, Anhui University, Hefei 230039, China
\* Correspondence: wjn@ahau.edu.cn

**Abstract:** Some policies implemented during the pandemic extended the time that students spend on electronic devices, increasing the risk of physical and eye strain. However, the role of different environments on eye strain recovery has not been determined. We recruited 20 undergraduate students (10 males and 10 females) from a university in eastern China and explored the restoration effects of their eye strain in different types of spaces (wayside greenspace, a playground, a square, and woodland) on campus through scale measurements. The results showed that the eye strain of the students accumulated by 15 min of e-learning was significantly relieved after 10 min of greenspace exposure compared to the indoor environment, and the recovery effect varied depending on the type of landscape. The effect of eye strain relief was found to be positively correlated with temperature, wind speed, visible sky ratio, canopy density, tree density, and solar radiation intensity, while it was negatively correlated with relative humidity. These findings enrich the research on the restoration benefits of greenspaces and provide a basis for predicting the effect of different environments on the relief of eye strain.

**Keywords:** campus landscape; greenspace; COVID-19; eye strain; restorative effects

## 1. Introduction

To block the spread of the COVID-19 disease, governments around the world carried out a series of different public health measures such as social distancing, quarantine, lockdowns, and curfews [1,2] while appealing to their citizens to avoid non-essential activities. These policies reduced outdoor activities [3] and increased the risk of smartphone overuse [4]; screen time rose by 400 min per week, as reported in [5]. Some new terms such as "videoconference fatigue" or "Zoom fatigue" became popular among the crowd [6]. The excessive usage of electronic products may cause psychological problems such as depression and/or anxiety, which, in turn, can lead to sleep disorders [7]. Furthermore, it can also result in problems with vision and hearing [8] and different skeletal muscle pains such as neck, lower back, and shoulder aches [9]. Among these symptoms, eye strain is the most common, which is characterized by dry, itchy eyes, foreign body sensations, tearing, and blurred vision [10].

Since the biophilia hypothesis [11] was proposed, health benefits from nature exposure have drawn increasing attention. The concepts of forest bathing (Shinrin-yoku) and forest therapy have been put forward one after another, and studies on this theme have expanded substantially over recent years [12]. Several studies have shown that, compared to the built environment, forest landscapes or greenspace can directly or indirectly relieve mental stress [13–15], restore physical health [16–18], enhance immunity [19], and reduce the inflammatory response [20,21], among other things. Since tertiary students living on

campus were unable to freely enter or leave school during pandemic-related isolation, the landscape on campus has become an alternative to parks or urban greenspace.

A network or system that consists of woodlands, tree groups, and individual trees located in an urban area can be considered an urban forest [22]. Like the forest environment, access to the greenspace on campus has proven to be beneficial. As a type of urban forest, it is associated with attention restoration [23,24] and the promotion of positive moods [25–27]. Some recent studies utilized virtual reality (VR) technology to explore the impact of different campus landscapes on college students' physiological rehabilitation and/or psychological recovery. Wang et al. argued that campus greenspace ameliorates the physical and mental fatigue conditions of college students and improves their concentration; among four types of greenspace, woodland and water landscapes have the best mitigation effect [28]. Ha and Kim found that a high-biodiversity landscape with natural sound contributes to the benefits for students' restoration [29]. Gao et al. suggested that the naturalness of the greenspace landscape has a significant impact on physical and psychological recovery [30].

Despite some studies that have shown that simulated nature has a positive impact on the physiological and psychological effects of subjects [31], with these effects being almost the same as those of the natural environment [32], VR cannot reproduce a complete real environment or the length of exposure to nature that people often experience. In addition, it may cause discomfort, especially in terms of motion sickness and eye strain symptoms [33–35]. Thus, there is still a gap in field research on the recreational effects of eye strain in different campus spaces.

Our study recruited college students who were in quarantine during the COVID-19 pandemic. The aim was to determine whether different greenspaces on campus have different effects on the participants' relief from eye strain. This research is hoped to supplement the theory of campus healthcare landscapes and provide a scientific basis for its future design and construction.

## 2. Materials and Methods

### 2.1. Study Sites

This on-site experiment was carried out at Anhui Agricultural University (AAU). AAU is located in the center of Hefei city, eastern China, covering an area of 212 hectares. As the capital of Anhui province, Hefei has developed rapidly in recent years, with the population increasing from 5.702 million in 2010 to 9.37 million in 2020 [36]. The annual temperature is 15.7 °C, while the average relative humidity (RH) is 77%.

During our experiment (May to June 2022), four types of different campus spaces (wayside greenspace, a playground, a square, woodland) (Figure 1) were selected as the study sites, and a classroom was used for the control group. The environmental characteristics of the four sample plots and the classroom control are shown in Table 1.

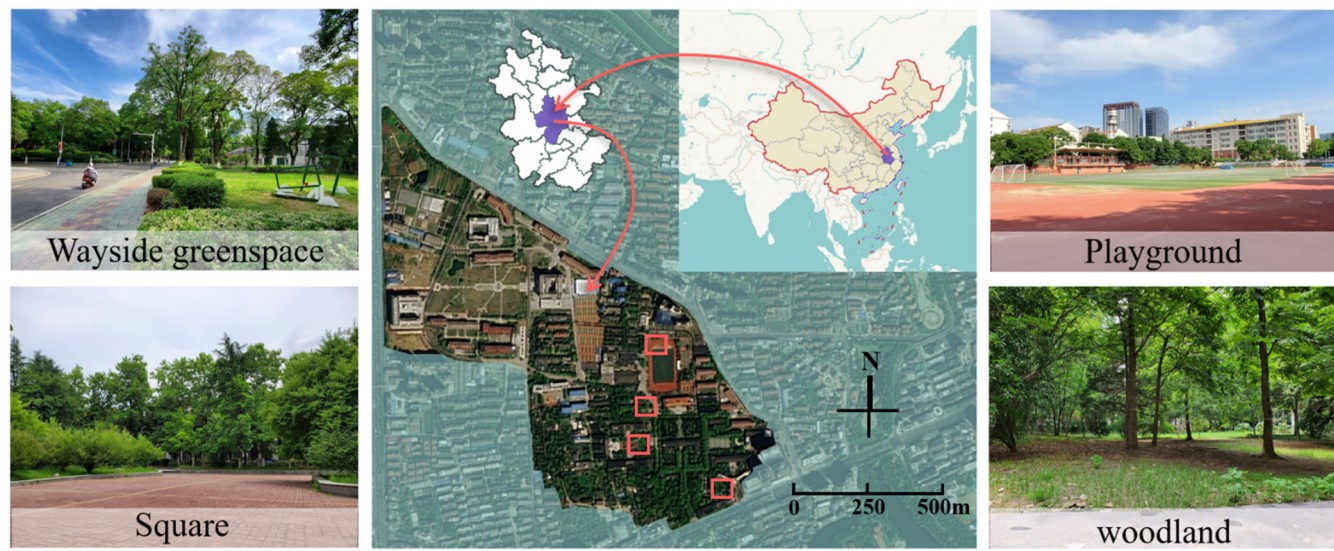

**Figure 1.** Experimental site and the scenery of different landscapes on campus.

**Table 1.** The environmental elements of the four sample plots and the classroom control.

| Environmental Factors | Wayside | Playground | Square | Woodland | Classroom |
|---|---|---|---|---|---|
| Temperature (Mean) | 29.85 °C | 28.12 °C | 32 °C | 27.6 °C | 25.9 °C |
| Relative humidity (Mean) | 42% | 46% | 40.05% | 56.45% | 59% |
| Wind speed (Mean) | 1.2 m/s | 1.78 m/s | 0.9 m/s | 0.6 m/s | 0 m/s |
| Light intensity (Mean) | 12740 lux | 61675 lux | 8820 lux | 6020 lux | 233 lux |
| Green view index | 86.7% | 14.17% | 54.33% | 93.03% | - |
| Visible sky ratio | 10.6% | 40.25% | 30.59% | 5% | - |
| Canopy density | 0.33 | 0.1 | 0.6 | 0.95 | - |
| Tree density | 0.012/m² | 0.00425/m² | 0.043/m² | 0.034/m² | - |
| Plant species | 18 | 12 | 16 | 30 | - |
| DBH [1] (Mean) | 42.2 cm | 48.2 cm | 30.2 cm | 31.4 cm | - |
| Size | 9850 m² | 18500 m² | 7500 m² | 3590 m² | 56 m² |

[1] DBH: diameter at breast height.

### 2.2. Participants

Twenty healthy college students (male/female = 1:1; Table 2) who always live in their dormitories at school were recruited. None of them had a high blood pressure, heart disease, eye damage, or brain defects, nor did they smoke or drink alcohol. All candidates were informed about the trial procedure and the purpose of the experiment before being selected as volunteers. They all agreed that their data could be used and that they would follow requests at any time during the period of our investigation. We made sure that their privacy and human rights are protected.

**Table 2.** Information about study subjects.

| Parameter | Value |
|---|---|
| Total | 20 |
| Major | Landscape architecture |
| Male number | 10 |
| Age | 20.75 ± 1.87 |
| Height (m) | 1.71 ± 0.09 |
| Weight (kg) | 65.15 ± 13.07 |
| Body mass index | 22.09 ± 3.67 |

### 2.3. Experimental Design

To avoid interference between people, only one male and one female participant were invited for a trial at the same time. Before the start of the test, the subjects were informed of the experimental process and relevant precautions in advance. After that, they were asked to fill in a demographic form and stay sedentary for 2 min; then, their initial eye strain level was examined. Next, the subjects were asked to learn a 10 min online video course in a tent (simulating an indoor learning environment) next to the selected sample plot and to complete the corresponding exercises on a tablet within the following 5 min. After completing this stress stimulation task, they were required to fill in the eye strain scale. After that, all of the participants were instructed to enter the specific campus space to perceive on their own for 10 min. We encouraged them to experience the natural environment with multiple senses. At the end of the experiment, their eye strain condition was tested again (Figure 2).

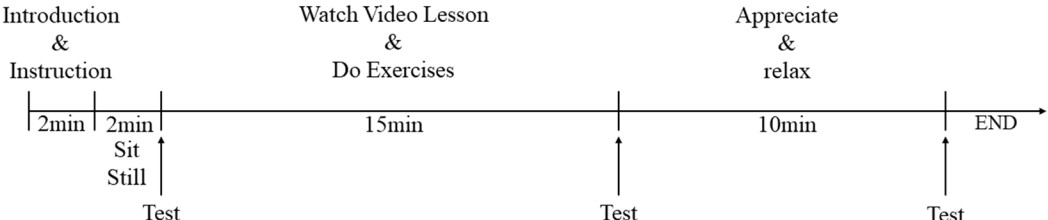

**Figure 2.** The procedure of the experiment.

### 2.4. Data Collection

#### 2.4.1. Eye Strain Level

A widely cited Hayes scale [37] was administered to estimate the degree of eye strain of the participants, comprising 10 items, including "To what extent do you experience: blurred vision at near distances/blurred vision at intermediate distances/blurred vision at far distances/difficulty or slowness in refocusing my eyes from one distance to another/irritated or burning eyes/dry eyes/eyestrain/headache/tired eyes/sensitive to bright light?" Each question was measured on a 7-point scale (0 = None; 1 = Slight; 2 = Mild; 3 = Moderate; 4 = Somewhat bad; 5 = Bad; 6 = Severe). The higher the grade, the worse the eye strain condition.

#### 2.4.2. Environmental Elements

We counted the number and variety of plants in each field. The DBH and canopy density of each tree were also measured, which may indicate their age and size. Temperature, relative humidity, and wind speed were monitored by a mobile weather meter (Kestrel 5000, Nielsen Kellerman Co., Boothwyn, PA, USA), while light intensity was measured by a portable luminometer (TA8123, TASI Electronics Co., Suzhou, China).

The green view index and visible sky ratio were computed by a human–machine adversarial scoring framework [38] through photos of the sites. The participants' preference of the sites was determined by their rating for each site, with scores ranging from –3 to 3, representing "dislike very much" to "very fond of."

### 2.5. Statistical Analysis

The data obtained from our trials were imported into Origin 2022 (OriginLab Corp., Northampton, MA, USA) for analysis and graphing. A paired sample *t*-test was chosen to compare the difference in the subjects' eye strain levels between post-stimulus and post-relaxation in each space. Differences in the recovery effect of eye strain by the spaces were tested through a one-way ANOVA. If the value showed a significant difference between each plot, a further pairwise comparison was made using the Fisher's least significant

difference (LSD) method. Furthermore, we used Spearman's correlation to investigate the link between environmental factors and the restorative function of campus greenspace.

### 3. Results

*3.1. Effects of the Campus Environment Intervention on Eye Strain*

The university students' levels of eye strain showed an overall increase after completing the 15 min strain stimulus task, while all exhibited relief after 10 min of campus greenspace exposure (Figure 3). The results of the paired sample *t*-test (Table 3) revealed that the subjects presented significant differences before and after the relief of eye strain in the playground, wayside greenspace, woodland, and square ($p < 0.05$). In contrast, this difference was not significant in the classroom setting ($p > 0.05$).

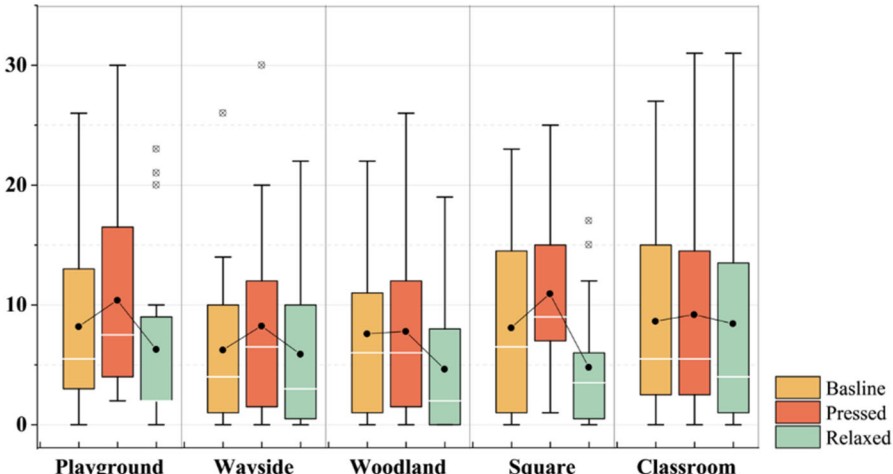

**Figure 3.** Eye strain level of baseline, post-stress, and post-relaxation in different settings.

**Table 3.** T-test results between post-stimulus and post-relaxation.

| Plot | Baseline | Post-Stimulus | Post-Relax | *p*-Value |
|---|---|---|---|---|
| Playground | 8.15 ± 7.52 | 10.35 ± 8.50 | 6.25 ± 7.22 | 0.001 *** |
| Wayside greenspace | 6.20 ± 6.44 | 8.20 ± 8.15 | 5.85 ± 6.45 | 0.001 *** |
| Woodland | 7.55 ± 7.00 | 7.75 ± 7.67 | 4.60 ± 5.69 | 0.000 *** |
| Square | 8.05 ± 7.39 | 10.90 ± 6.96 | 4.75 ± 5.06 | 0.000 *** |
| Classroom | 8.60 ± 8.16 | 9.15 ± 8.57 | 8.40 ± 9.07 | 0.083 |

*** $p < =0.001$.

*3.2. Differences in the Relaxation Effects among Campus Spaces*

The results of the one-way ANOVA indicated that the eye strain recovery levels differed significantly across the five environments. Figure 4 shows the extent to which the university students recovered from eye strain in different campus spaces, i.e., the eye strain values measured after the stress task minus the values measured after greenspace exposure. It can be seen that the participants' eye strain recovery level in the square, playground, and woodland environments was significantly different from that in the classroom setting. The square was the most beneficial eye strain relief environment, which had a markedly different effect compared to the woodland and wayside greenspace.

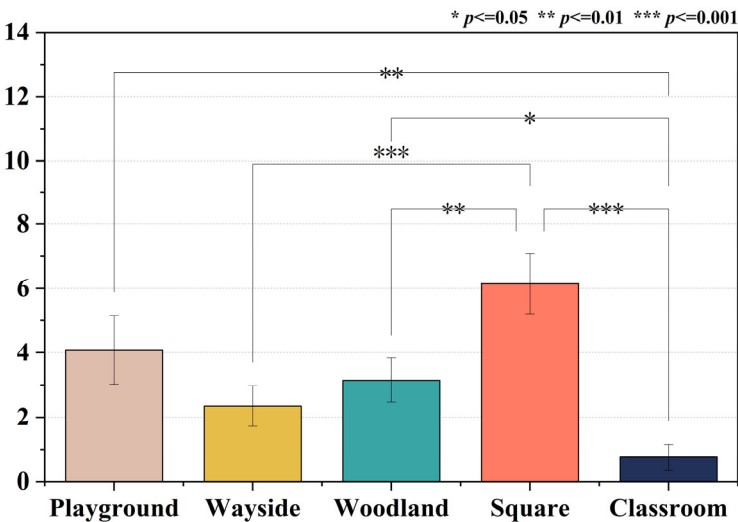

**Figure 4.** Comparison of the recovery degree from eye strain in different spaces.

### 3.3. Correlation between Environmental Factors and Eye Strain Recovery

Of all the environmental factors we studied, relative humidity (RH; $R = -0.347$, $p < 0.001$) showed a significant negative correlation with the degree of recovery from eye strain (ESR). The visible sky ratio (VSR; $R = 0.348$, $p < 0.001$), light intensity (LI; $R = 0.205$, $p < 0.05$), temperature (Temp; $R = 0.347$, $p < 0.001$), wind speed (WS; $R = 0.205$, $p < 0.05$), canopy density (CD; $R = 0.289$, $p < 0.01$), and tree density (TD; $R = 0.393$, $p < 0.001$) were significantly positively correlated with the restoration effect, while the green view index (GVI; $R = 0.146$, $p > 0.05$), preference (Pref; $R = 0.163$, $p > 0.05$), plant species (PS; $R = 0.107$, $p > 0.05$), and BDH ($R = 0.101$, $p > 0.05$) demonstrated no relationship with eye strain relief (Figure 5).

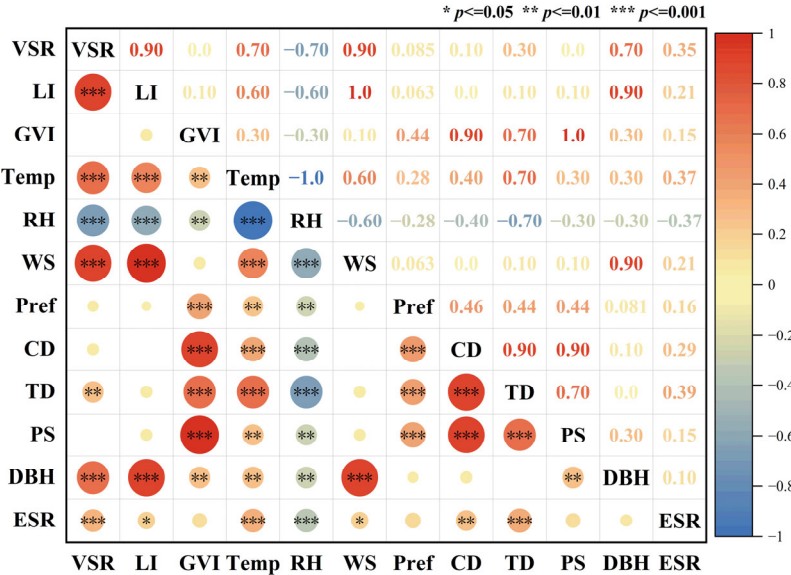

**Figure 5.** Spearman correlation matrix showing the relationships between environmental factors and eye strain recovery. Red and blue colors denote positive and negative correlations, respectively; intensity of color reflects the strength of the relationship.

## 4. Discussion

### 4.1. Impact of Campus Greenspace on Eye Strain Restoration

Visual display terminals (VDTs) such as TVs, computers, smartphones, and tablets have become an integral part of daily life [39], especially after the occurrence of global

public health emergencies. Several studies conducted during the COVID-19 pandemic have shown an increased onset and exacerbation of eye strain [40,41], dry eye disease [42], and myopia [43,44] from electronic device use. Our study also indicated that 15 min of e-learning added to the risk of eye strain in most students. A stroboscopic effect [45], inappropriate brightness [46], screen glare [47,48], polarized light [49,50] from VDTs, and prolonged closed eye use [51,52] are possible causes of eye strain. At the same time, studying online courses requires the eyes to be constantly glued to relatively small words and pictures, which need excessive concentration, leading to fewer blinks, thus causing dry eyes [53–55].

A large amount of existing research suggests that nature exposure has many physiological benefits for people, including decreased obesity [56], improved sleep quality [57], and a reduced risk of cardiovascular disease [58] and even cancer [59]. Our findings revealed that greenspace is also good for ocular health, as 10 min of exposure provides relief from eye discomfort, while indoor environments do not have such a significant effect. These results are complementary to the existing health benefits of greenspace.

According to a report in 2014, approximately 23% of school-aged children, 64%–90% of computer users, and 71.3% of dry eye patients in China have varying degrees of eye strain symptoms [60]. Thus, there is a need to explore the possibility of rehabilitating patients with eye strain or dry eyes through low-cost forest bathing programs in the future.

### 4.2. The Influence of Environmental Factors on the Effectiveness of Eye Strain Recovery

A higher sky ratio means a more open field of vision, which may be more conducive to ciliary muscle relaxation [61]. Natural light exposure has been shown to be associated with dopamine secretion in the retina [62,63], and the light intensity outdoors is usually 10–1000 times stronger than that indoors [64]. This may explain why open-air environments have a better relaxing effect on eye strain compared to indoor environments.

The tear film is composed of mucin, aqueous, and a lipid layer [65], which is responsible for moisturizing the eye and allowing for clear vision [66]. Temperature showed a positive correlation with eye strain recovery, probably because the elevated temperatures within a certain range can facilitate lipid melting in the meibomian gland [67,68]. However, contrary to common assumptions, our results suggested a negative correlation between the degree of eye strain recovery and relative humidity but a positive correlation with wind speed. Increased humidity is generally considered to improve tear film stability, while higher wind speeds can lead to intensified tear evaporation [69,70]. The opposite outcomes of our study may be explained by the fact that excessive humidity might cause discomfort [71], while stronger winds in summer may increase one's thermal comfort. However, the reason still needs to be further investigated.

In addition, one study concluded that the visual perception of humans is more pleasant when the green view index is above 25% [72], but our study did not find an association between GVI and eye strain recovery. Similarly, some studies have pointed out that biodiversity is beneficial for mental health and wellbeing [29,73–76], while our results did not show a correlation between the degree of eye strain recovery and biodiversity, just as Chang et al. demonstrated that physiological responses basically remain unchanged when biodiversity increases [77].

These findings allow us to predict the effect of different environments on the healing of eye strain to some extent in the future. Notably, greenspace has alleviating effects on the psychological pressure raised during the pandemic [78–81], and there might be a synergistic effect between the different health benefits of greenspace [81]. This may have led to a reduction in the students' stress from studying online courses and unconsciously increased the frequency of eye blinks of the subjects because their attention was restored, resulting in lower reported eye strain. Furthermore, the relationship between nature doses and health benefits is complex [82], and the impact of nature on a human being can be instant or postponed and may even fade over time [83]. Some studies have suggested that a higher dose and a longer duration can lead to greater gains [84–86], but there are no

further gains after the positive association reaches its peak [87]. Therefore, despite our results supporting the idea that simply spending a brief amount of time in a forest environment can be physically relaxing [18], the duration of exposure may cause varying outcomes.

### 4.3. Limitations

To the best of our knowledge, this is the first study to explore the restoration effect of the greenspace environment on eye strain, and there are several limitations to this study that are worth noting. First of all, the sample size of the experiment ($n$ = 20) was relatively small. Although many studies (e.g., [18,71,88,89]) on the health benefits of forests have used 10–20 volunteers as subjects, and An et al. claimed that a sample of three would satisfy the high power (90%) of the theoretical sample size [71], the larger the sample size, the more convincing the results, statistically. Second, the self-evaluated eye strain of those students who major in landscape architecture may not reflect their actual physiological condition, since many studies showed that the well-being benefits of greenspace are related to individual perceptions [90–92]. Thus, subsequent studies could recruit a more universal experimental group or use an electrooculogram [93], regional brain wave monitor [94], eye tracker [95], and other objective assessment approaches as a substitution for the scale. Third, only healthy, young university students were selected as the subjects; future research into the therapeutic effects of school landscapes could be extended to people of all ages or those with dry eyes. Fourth, the durations of both eye strain and greenspace exposure employed in this study were relatively short, and the effects of different exposure durations could be further investigated. Moreover, the variables could not be fully controlled as a field experiment. Meanwhile, the green view index and visible sky ratio calculated through photographs may not be consistent with human perception in a real environment.

### 5. Conclusions

Though this on-site experiment was a preliminary exploration of the effects of different campus landscapes on the recovery of students' eye strain, it provides new insight into the relationship between greenspace and human health. Through an exploration of 20 university students, we found that 15 min of online learning can increase students' risk of eye strain, while 10 min of campus greenspace exposure can provide effective relief from accumulated eye strain, and subjects even feel better than they did initially. The degree of restoration was positively correlated with the temperature, wind speed, visible sky ratio, canopy density, tree density, and solar radiation intensity of the site, while it was negatively correlated with relative humidity. In terms of the results, we recommend that students get in touch with nature to relieve accumulated eye strain after studying online courses or follow the 20–20–20 principle of eye use (after looking at a phone or computer for 20 min, look up 20 feet away for at least 20 s to relax) if nature contact is not available.

**Author Contributions:** Conceptualization, J.W. and Y.L.; methodology, Y.L.; formal analysis, Y.L.; investigation, Y.L., J.C., and Y.Y.; resources, J.W.; writing—original draft preparation, Y.L.; writing—review and editing, J.W. and H.Z.; visualization, Y.L.; supervision, J.W., B.Z., H.M., and T.H.; project administration, J.W and Y.L. All authors have read and agreed to the published version of the manuscript.

**Funding:** This research received no external funding.

**Institutional Review Board Statement:** The study was conducted according to the guidelines of the Declaration of Helsinki and under the supervision of the Ethics Committee of Anhui University.

**Data Availability Statement:** The data presented in this study are available on request from the corresponding author.

**Acknowledgments:** We are grateful to the reviewers for their professional comments, which greatly improved this paper. We thank Yitao He, Xianle Bian, Zi Wang, Qing Deng, and all the other volunteers who have contributed to this research.

**Conflicts of Interest:** The authors declare no conflict of interest.

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
