# Peer review of "Impact of Environmental Factors on Short-Term Eye Strain Relief during COVID-19 Quarantine: A Pilot Study"

_forests, doi:10.3390/f13111966_

Round 1
Reviewer 1 Report
Introduction: When talking about Forest bathing you could mention that it was introduced in Japan to combat ‘techno-stress’ i.e. too much screen-time (Qing Li book reference).
Method: Line 118 what is DBH?
Is 15mins considered enough time to create eye-strain? Could you provide rationale for why 15mins was chosen?
Discussion: Could you talk a little more about the proposed mechanisms for why viewing green spaces might alleviate eye-strain? For example, going back to the biophilia hypothesis you could talk about attention restoration and fractals i.e. how viewing natural geometric patterns and familiar soothing green colours might alleviate eye-strain. Anything about long-distance viewing or peripheral vision?
Author Response
We are very grateful for your patience and your useful comments on the manuscripts, which are important to us.
Point 1: Introduction: When talking about Forest bathing you could mention that it was introduced in Japan to combat ‘techno-stress’ i.e. too much screen-time (Qing Li book reference).
Response 1: Could you please tell us the name of the reference? We couldn't find it.
Point 2: Method: Line 118 what is DBH?
Response 2: DBH has been explained in the footnotes to Table 1.
Point 3: Is 15mins considered enough time to create eye-strain? Could you provide rationale for why 15mins was chosen?
Response 3: In pre-experiments, 15 minutes of online learning tasks were adequate to provoke visual fatigue in most people. Of course, the level of fatigue increased with time. However, longer times can cause negative emotions in the subjects, which may lead to impatience and thus affect the cooperation with the experiment. The title of this article is now changed to: Impacts of Environmental Factors on Short-term Eye Strain Relief During COVID-19 Quarantine: A Pilot Study
Point 4: Discussion: Could you talk a little more about the proposed mechanisms for why viewing green spaces might alleviate eye-strain? For example, going back to the biophilia hypothesis you could talk about attention restoration and fractals i.e. how viewing natural geometric patterns and familiar soothing green colours might alleviate eye-strain. Anything about long-distance viewing or peripheral vision?
Response 4: Relevant discussion has been added at the end of section 4.2. now. Our study is focusing on the relationship between limited environmental factors and eye strain, there are really too many possible influencing factors.
Reviewer 2 Report
Thank your for this interesting paper.
This study could have been improved when conducting a sample size calculation to become a more sound statistical evidence.
Please add in Table 3 and Figure 4 the appropriate labelling. Which type of values is displayed should please be indicated in the legend.
Additionally, please add to Table 3 the baseline values for all different sites.
Author Response
Dear referee,
Thank you very much for the thorough and helpful review of the manuscript. By changing the text according to this response, we hope to improve the manuscript and to account for the concerns you have raised.
Point 1: This study could have been improved when conducting a sample size calculation to become a more sound statistical evidence.
Response 1: That’s true. We have also taken this into consideration. However, constrained by various factors, we have done this small-scale pilot study to explore whether further research is worthwhile. We have referred to other similar studies in determining the sample size. For example, there were 12 participants in Lee et al. (2009), Park et al. (2010, 2011), Song et al. (2013a), Tsunetsugu et al. (2013), 13 participants in Song et al. (2013b), 14 participants in Duncan et al. (2014), and 15 participants in Horiuchi et al. (2014). This has been discussed in 4.3. Limitations.
Point 2: Please add in Table 3 and Figure 4 the appropriate labeling. Which type of values is displayed should please be indicated in the legend.
Response 2: Inappropriate labeling due to inadvertence has been modified. The horizontal axis of Figure 4 is marked with the data type under.
Point 3: Please add to Table 3 the baseline values for all different sites.
Response 3: Baseline values now have been added.

Reviewer 3 Report
This is a potentially interesting and useful study. It has a clear aim, has been in general carefully carried out and is well-written and presented. I commend the authors on their careful presentation and writing in a second language (I assume) - only minor revision of the odd phrase is required. It is accurately described as a pilot study which could be expanded to yield more interesting and robust results. However, even as a pilot study, there seem to be some fundamental flaws in the way the study was carried out and the results presented and discussed.
I am not recommending rejection, and I would like to see this study published when revised, but I consider there are several weaknesses that must be addressed before acceptance. See detailed comments below.
Line 3: recommend the title should read: "Impacts of Environmental Factors on Short-term Eye Strain..." (see below)
Line 28: reference 2 is not closely related to the statement made.
Lines 43-45: Helpful to clarify that this study relates to tertiary students living on campus. The situation for school students living at home, also subject to on-line learning and activities but without the campus amenities described in the study, would be quite different.
Lines 49-58: The material on VR here is quite confusing and maybe redundant. In particular VR would not normally simulate the lengths of exposure to nature that people often experience. As I argue later, one of the main weaknesses of this study is that the exposures studied are so short. There are studies suggesting that longer-term exposure may yield different results to those obtained in this study.
Figure 1 and Table 1: Please check the DBH results presented. These seem to be very large. Possibly circumference measurements have been used in error? If they are correct, this shows a weakness in measuring only 4 sample plots - the results in Table 1 do not illustrate the differences in these types of green space (GS) that I would expect. I'd be interested to know the age of the vegetation in particular the woodland plot.
Line 80: only 4 sample plots shown in table 1 - not the classroom control.
Line 91: I hope "always under-protected" is a typo and that you mean "privacy and human rights are protected"
Table 2: Selecting only Landscape Architecture students is a potential weakness for this study, as many of the reported benefits of GS exposure relate to perceptions of that exposure. LA students might be expected to have a greater than average existing knowledge of these benefits and understanding of the purpose of the research (even if this was not explained prior). I recommend that this is discussed in section 4.3, and as this is stated to be a pilot study, the authors recruit a more representative student sample in later investigations.
Lines 94-95. I do not understand this sentence. What are the "single trials" and how do these contribute to or differ from the results presented which I would have thought come from testing of each student individually.
Figure 2: I accept the basis of the procedure selected, but it would be interesting to see further studies undertaken on a different timeline whereby "appreciate and relax" preceded rather than, or as well as, the on-line exposure. See later comments on Fig 3.
Fig 3 and Table 3: potentially very interesting results, but their significance may be diminished because of the difference in pre-stimulus test results between the different environments, which are not presented but appear to be significant. In particular why are the classroom first test results so much higher than all the others? If the authors cannot explain this result then many of the subsequent results may be meaningless.
Lines 155-166 and Fig 5: Quite difficult to understand and interpret. Fig. 5 seems to show a 2-way analysis but this is not referred to in the preceding paragraph, nor discussed in section 4.
Section 4: Somewhat brief superficial in light of the potentially interesting results. Refer to previous comment about Fig 5.
Lines 180-183: In addition to reference 58, earlier references cited in the Introduction should be cited and results compared with those in the current study.
Section 4.2. Good discussion of the results in terms of physical wellbeing characteristics, but much weaker in terms of mental wellbeing characteristics and your results in terms of people's perceptions of the environment (both of which are referred to in the Introduction and in the psychological studies cited). In particular the authors must discuss the limitations of the very short duration of both eye-strain and GS exposure used in this study. There is ample literature now available on the effects of GS exposure "dose" - duration, intensity etc. See works of e.g. Fuller, Mitchell, Shanahan, Lin etc.
Section 4.3. This section must be expanded to take account of comments above including: small sample size, short duration of exposures*, narrow range of study subjects, fixed order of exposure and testing, variable baseline test results between the different environments and control* (* referring to the one I consider the most critical).
Round 2
Reviewer 3 Report
Thanks to the authors for their careful and respectful responses to my comments. I am glad that you appeared to find my comments helpful.
I have no hesitation in recommending the revised paper for publication subject to relatively minor further revision. In respect of your detailed responses, most of them are very good and have cleared up the points raised by me. It is somewhat laborious for me to respond to them individually so I have only commented on those where I recommend you give some further attention and consider minor revision.
Response 4: I accept your points about VR and its weaknesses. But I don't think these concerns are well expressed in lines 59-64. In particular I refer to my original comment that "VR would not normally simulate the lengths of exposure to nature that people often experience" and suggest you specifically refer to this weakness. It is, after all, related to the revised title of the article. In general, I would still like to see more reference to the duration of exposure.
Response 5: I am familiar and quite comfortable with the practice of using canopy density and DBH as a proxy for age, but I suggest that you refer to this rationale briefly but specifically, within section 2.4.2.
Response 15: As mentioned I would still like to see more reference to the dose of exposure, in particular the duration. Response 15 was strong and effective as a discussion of your results. I would strongly suggest that you use some of the material in this response to expand your new lines 234-238 to discuss the duration of response more fully including perhaps citation of some of the other references (Hartig et al, Fuller et al).
Response 16: Testing only short-term responses is not necessarily a weakness of your study (it was probably a necessary consequence of your experimental design) but it remains a limitation and I strongly recommend that this is stated in Section 4.3, probably with reference to some of the studies cited in response 15.
Author Response
Dear Professor,
We are very grateful for your support and further suggestions. You are so kind. We hope that you will be satisfied with this revision.
Point 4: I accept your points about VR and its weaknesses. But I don't think these concerns are well expressed in lines 59-64. In particular I refer to my original comment that "VR would not normally simulate the lengths of exposure to nature that people often experience" and suggest you specifically refer to this weakness. It is, after all, related to the revised title of the article. In general, I would still like to see more reference to the duration of exposure.
Response 4: Your concern has been added in lines 63-64 :
“VR could not reproduce the complete real environment and the lengths of exposure to nature that people often experience.”
Point 5: I am familiar and quite comfortable with the practice of using canopy density and DBH as a proxy for age, but I suggest that you refer to this rationale briefly but specifically, within section 2.4.2.
Response 5: This rationale has now been mentioned in section 2.4.2. :
“The DBH and canopy density of each tree was also measured, which may infer their age and size.”
Point 15: As mentioned I would still like to see more reference to the dose of exposure, in particular the duration. Response 15 was strong and effective as a discussion of your results. I would strongly suggest that you use some of the material in this response to expand your new lines 234-238 to discuss the duration of response more fully including perhaps citation of some of the other references (Hartig et al, Fuller et al).
Response 15: The discussion of duration is expanded with more references cited :
“Furthermore, the relationship between natural dose and health benefits is complex [83], the impacts of nature on a human being can be instant or postponed, or they may fade over time [84]. Some studies suggest that higher dose and longer duration could lead to greater gains [85-87], but there were no further gains after the positive association reached its peak [88]. ”
Point 16: Testing only short-term responses is not necessarily a weakness of your study (it was probably a necessary consequence of your experimental design) but it remains a limitation and I strongly recommend that this is stated in Section 4.3, probably with reference to some of the studies cited in response 15.
Response 16: This limitation has been stated in Section 4.3. :
“Fourth, the duration of both eyestrain and green space exposure employed in this study was relatively short, and the effects of different exposure durations could be further investigated.”